# Living with vaccine-induced immune thrombocytopenia and thrombosis: a qualitative study

Paul Bennett  ,[1] Filiz Celik,[1] Jenna Winstanley,[1] Beverley J Hunt,[2] Sue Pavord[3]

¹School of Psychology, Swansea University, Swansea, UK
²Guy's and St Thomas' NHS Foundation Trust, London, UK
³Oxford University Hospitals NHS Foundation Trust, Oxford, UK

**Correspondence to**
Professor Paul Bennett;
p.d.bennett@swansea.ac.uk

## ABSTRACT

**Objectives** To explore the experiences of people up to 18 months after being diagnosed with vaccine-induced immune thrombocytopenia and thrombosis (VITT).

**Design** A semistructured qualitative study, conducted via Zoom, of a cohort of people with VITT.

**Setting** Participants discussed their experiences of hospitalisation and following discharge.

**Participants** 14 individuals diagnosed with VITT, recruited via a Facebook support group and advertising on Twitter.

**Results** Thematic analysis identified challenges of obtaining medical care and diagnosis; fear of the severity of symptoms and unclear prognosis; and lack of family support due to isolation imposed by the COVID-19 pandemic. Once home, participants experienced continued significant symptoms; fear of recurrence; inadequate medical knowledge of their condition; and difficulties coping with residual physical disabilities and psychosocial losses. Also reported were feelings of isolation and abandonment due to lack of government support.

**Conclusions** This is a significantly challenged group of people, with multiple health, financial, social and psychological losses. These losses have been compounded by experiences of limited governmental and societal recognition of the problems they face.

## INTRODUCTION

Following the first waves of the COVID-19 pandemic in 2020, governments across the world instigated a range of infection control measures including mass vaccination. Although effective in reducing the severity of COVID-19 infection, a very small percentage of individuals receiving adenoviral vector-based vaccines experienced vaccine-induced immune thrombocytopenia and thrombosis (VITT),[1] which involved life-threatening thromboses in multiple body sites. At onset, symptoms of thrombosis included severe pain (unremitting headache, backache, abdominal pain, chest pain), typical manifestations of deep vein thrombosis or pulmonary embolism (PE), such as leg swelling and breathlessness and in extreme cases seizures and disturbed consciousness. Intracerebral bleeding occurred in approximately one-third of those with cerebral vein thrombosis, with

high mortality risk.[1] Survivors report a range of long-term debilitating symptoms including those related to stroke (eg, limb weakness or paralysis, dysphasia, chronic headache), PE, myocardial infarction, hepatic hypertension, loss of limb and chronic fatigue.

The need for long-term psychological care has also been acknowledged.[2] This should not be surprising. The sudden-onset, life-threatening and life-changing nature of VITT would predict high levels of associated psychological distress among survivors, similar if not greater than those found in patients with more 'classic' thrombosis.[3 4] However, additional factors may increase this risk. First, they were the consequence of a deliberate choice by the individual to receive the vaccination for either personal or public health purposes: choices that were encouraged by the government and medical authorities. Second, VITT was a completely new syndrome lacking previous medical experience. Third, survivors typically experienced severe symptoms and required intensive medical care at a time when hospital visiting was not allowed due to COVID-19 restrictions and consequently, they may have experienced significant psychological trauma[5] with little or no immediate social or family support. An important moderating factor is likely to be the degree of family and societal support available to those with VITT once discharged.

This study aimed to explore patients' experiences in hospital and after discharge, using interviews with people diagnosed with VITT, conducted up to 18 months following the acute phase of their disorder.

## METHOD
### Patient and public involvement
The protocol of this study and likely impact of publication were discussed separately with a number of patients with VITT prior to conducting the research. In addition, all participants had sight of the paper as it developed (on two occasions) to ensure their agreement with the themes identified, quotations cited and degree of anonymity provided.

### Recruitment and sample
Participants with a confirmed diagnosis of VITT were recruited via a number of social media routes: (1) a closed VITT Facebook support group supported by Thrombosis UK, (2) the Thrombosis UK Twitter feed, (3) and retweets by individuals known to members of the Facebook group or who read the Twitter feed. As a veracity check, participants were asked to confirm the date of their diagnosis and the hospital in which the diagnosis was made. In addition, the interview included a description of their initial symptoms which were checked for consistency with those of VITT.

The final sample comprised 14 participants, 10 of whom were participants in the support group and 4 recruited from the wider VITT population throughout the UK. All made direct contact with the research team using an email address provided. Due to the relatively small number of people with VITT, details of participants are summarised rather than detailed to facilitate confidentiality. The final sample comprised 11 women and 3 men, with a mean age of 43.7 years, with interviews conducted 14–18 months post-vaccination. All were white British (including one nationalised). All had jobs prior to receiving the vaccine; only three had fully returned to work at the time of interview and one person was actively seeking work. Prior to interview, using the UK Office of National Statistics Socio-Economic Classification,[6] one participant fell within the upper managerial category, seven in the lower managerial/administrative/professional boundary, three small employers, two routine/manual workers and one 'intermediate'. Although the sample size was determined by the number of people responding to the research advert, the themes achieved saturation.

### Method
The study interviews were conducted between August and November 2022. Participants who contacted the study team were sent a Participant Information Sheet providing details of the study and asked to affirm their willingness to participate by making further contact with the study lead. This also allowed them time to discuss any questions or issues concerning the research. After providing informed consent, participants took part in a semistructured virtual interview on Zoom, addressing their experience of hospitalisation and the longer-term impact of VITT. Three people conducted the interviews (PB, FC, JW). All are trained in clinical/counselling psychology and related interview techniques. PB and FC have significant experience in qualitative research interviews. JW had less experience, and the transcripts of her interviews were checked to ensure adherence to the semistructured interview protocol. PB had worked clinically with some patients with VITT. JW and FC had no previous knowledge of VITT. No interviewee was known to their interviewer. The interview schedule is available as an online supplemental file 1.

### Data analysis
Interviews were recorded on Zoom and a 'clean' transcript determined, after which, following ethical requirements, the original recordings were deleted. All possible identifying elements were removed from the transcripts, which were analysed by PB using inductive thematic analysis.[7] Audio recordings were listened to repeatedly to ensure accuracy and enable the identification and generation of relevant initial codes and textual units for features and patterns in the data. Extracts and phrases were used to identify potential themes, with relevant data ('quotes') gathered within identified themes. The data were systematically reviewed to ensure that a name, definition and exhaustive set of data were identified to support each category. The same unit of text could be included in more than one category. To ensure rigour and trustworthiness of recruitment, data collection and analysis, the process of saturation was adhered to. Themes and subthemes derived from the data were checked and validated by FC; any differences were resolved through discussion.

## RESULTS
Two broad thematic themes arose: (1) *the trauma of care*, including difficulties of accessing appropriate treatment during the acute phase, the fear experienced in hospital as a consequence of the severity of symptoms and unclear prognosis, and lack of family support due to isolation imposed by the COVID-19 pandemic; (2) *surviving the long haul*, including the struggle to optimise care once discharged from hospital, the continuity of significant symptoms, fear of recurrence, a lack of recognition of their condition, and multiple physical and psychosocial losses (see table 1).

### Theme 1: the trauma of care
This theme comprised three subthemes: (1) multiple attempts to access care, (2) fear versus feeling a fraud, (3) the need for a human touch.

#### Multiple attempts to access and maintain care
Prior to March 2021, VITT had not been identified and the key diagnostic features were not known. Many

**Table 1** The themes and subthemes identified from the participant interviews

| |
|---|
| **Theme 1: the trauma of care** |
| Multiple attempts to access and maintain care |
| Fear versus feeling a fraud |
| The need for a human touch |
| **Theme 2: surviving the long haul** |
| Moving away from safety |
| The struggle to optimise care |
| The continuity of significant symptoms |
| Lack of support and wider recognition of VITT |
| Multiple losses |
| VITT, vaccine-induced immune thrombocytopenia and thrombosis. |

participants developed VITT around this time and more benign diagnoses were presumed. As a consequence, many had difficulties in accessing urgent care following onset of severe symptoms; for example, headaches were frequently diagnosed as migraines and other symptoms received a variety of alternative diagnoses.

Many took steps before calling for medical aid or seeking acute care. When participants did try to access secondary care (often following advice from National Health Service (NHS) England's telephone service, NHS 111), this was not always easy. Very few reported urgent hospital admission with suspected VITT; the majority described repeated attempts to seek treatment. Of note is that those seen by general practitioners (GPs) or general or emergency physicians were more likely to be misdiagnosed, while haematologists, when eventually engaged in care, were more likely to diagnose VITT, particularly in the early days of knowledge of the diagnosis. A, for example, who developed VITT in January 2021, described how she was seen in the emergency department and reassured that 'Oh, everything's fine, it's not vaccine related … There was something that was not quite brilliant, but he basically said, it's fine. So, you can go home. Just take paracetamol and ride it out.' While leaving hospital, a haematologist, identifying abnormal blood results, called her back to hospital where she was immediately admitted.

Similarly, R was seen by her GP in March 2021, who reassured her that it was not VITT: 'No, its fake news, it's impossible. There's no chance you have. Honestly, it just can't happen….' Her headaches were not addressed, and her abdominal pain diagnosed as 'gas': 'And I was like, I've had a baby, I had a C-section and after a C-section you get tremendous gas … This is not gas!! So, he prescribed me a drug for gas that obviously didn't do anything – I didn't have gas!' The following day, she attended the emergency department, where she was again treated for 'gas' and discharged with treatment and follow-up by an oral antacid in 5 days. She reports being subsequently contacted by her emergency physician who said, "your platelets are really low, so just be careful. If you cut yourself, you might bleed to death." She was not admitted to hospital until 2 days later, when her practice nurse undertaking a routine smear noted her significant ill health.

### Fear versus feeling a fraud

Virtually, all participants experienced intense fear once admitted to hospital, either because of the severity of their symptoms and/or their doctors' uncertainty about their prognosis. Many expected to die. C encapsulated a frequently cited experience: 'I honestly thought I wasn't gonna make it through the night, it was probably the scariest time ever. Because they wasn't explaining it to me properly…. it was almost like the Haematologist was excited and I've never seen anybody so excited to be giving a diagnosis to somebody. He had two or three other doctors with him every time he came to see me. And it is all smiles (!) and, you know talking about me like I wasn't there, and it was, it was, so scary. And every time I asked them will I survive this, you know, could this kill me? They said, we don't really know, we don't know enough about it, but you're on the correct treatment.'

Sometimes, fear of dying was brought about by more subtle aspects of care. AM noted, for example, when his wife was allowed to visit him, 'I knew that they thought this could be end of life. Because that was the only time you was allowed visitors', and recalls, 'asking her to leave 'cause every time I just kept falling asleep and waking up and every time I woke up, she was at the end of my bed. I remember asking her to leave. And part of that was to be honest that, if I was going to die, I didn't want her to see me die.' By contrast, A was one of a minority of patients who, when put in a ward of seriously ill patients, many of whom appeared to be receiving terminal care, felt guilty and a 'fraud' for taking a bed for what she felt were relatively minor symptoms.

Once participants were receiving consistent care, although they were clearly anxious about their condition, they typically gained confidence in their physicians (although there were exceptions), and most viewed their relationship with their haematologist very positively, describing them as 'brilliant' and 'lovely' and willing to go beyond the norm to support them. Such rapport was facilitated by being taken seriously and listened to (A).

### The need for a human touch

Due to COVID-19 restrictions, participants received minimal visits from relatives and friends while in hospital, which proved challenging for both patients and their relatives. A number of participants noted that this was worse for their partners, as they were often in a situation with little meaningful knowledge of their partner's well-being (or otherwise) and suffered significant anxiety as a consequence. One participant (A) described: 'It was bad for me, but worse for my partner. I was aware of what was happening to me, but he worried because he was left in the dark.' One described her partner seeing her 'on drugs and incoherent' on a social media platform

and as a result becoming seriously concerned about her well-being with no way of obviating this. Sometimes, the human touch was provided by healthcare professionals: 'Like when you wake up in the middle of the night…. remember there was COVID… and I remember crying, going, 'I just need a human touch', and I remember her (a nurse in ICU [intensive care unit]) holding my hand and how much that meant to me' (P).

### Theme 2: surviving the long haul
#### Moving away from safety
Although their experiences in hospital were often traumatic, many participants felt the hospital became a safe place, where they could receive immediate treatment should their medical condition deteriorate which, given their hospital experiences, was often a realistic concern. Discharge had a dual impact; participants were pleased to be back home but felt removed from their safe environment. As AM noted, if there were concerns, investigations occurred within hours in hospital but took months once discharged. C noted that there was no point of call should her condition deteriorate. She was told she should go to Accident and Emergency. Given the difficulties C experienced receiving care during the acute stages, both she and her partner found discharge very concerning and put him, in particular, into a 'state of panic'.

#### The struggle to optimise care
Most participants found the care provided by their haematologist to be excellent. They were variously described as 'brilliant', 'lovely' and 'amazing'. Most communicated well with participants and made themselves available should they have any concerns both in hospital and following discharge. These positive experiences, however, were not always reflected in their relationships with doctors in other specialties. One GP, for example, whom the participant has subsequently changed, attributed their fatigue and headaches to depression and long COVID despite having letters from the hospital reporting a diagnosis of VITT and never having been found positive for COVID-19 infection. Another (C) reported that their neurologist denied her migraine and fatigue were related to VITT, stating 'chronic migraines due to analgesic addiction and need to come off all painkillers instantly and also have chronic fatigue syndrome, which has got nothing to do with VITT.' 'I literally walked out that appointment crying.'

Many relied on support from family members, both in the early stages following returning home and in the longer haul: 'At which point I couldn't walk more than about 100 metres but that's only if I held someone's hand. I couldn't make it up the stairs. So, I remember X holding my hand, and we do one step, and then I just have to wait, and then we do another step. So, it took me quite a few months, so at this point my parents moved in to be my carers…. I couldn't shower I couldn't stand for long periods of time. I couldn't make it from the kitchen to the living room. I couldn't get up from the bed if I

wanted to go to the bathroom, so it was many months of self-rehabilitation. And I can't imagine what it would be like for people who have no one… like my heart broke because for a family member to take you to the toilet is one thing - for a stranger to take is another thing' (V).

Once care was established, participants had multiple contacts with a range of health professionals, often several times a week over periods of several months. Many had to travel long distances to go to centres of expertise rather than their local hospital, which was disruptive of day-to-day life and the constant reminders of their condition further inhibited their psychological recovery. These visits were typically stressful as participants, at least initially, experienced some degree of traumatic memories visiting the hospital in which they had been so acutely ill, and worried about any results they may be given and/or the inherent unpleasantness of procedures such as endoscopy. Anxiety was not always just evident on the day of testing. F, for example, noted, 'When I go to my hospital appointments the anxiety is pretty much off the chart, it's like really… you know, it affects my bowels, it effects everything, just like, yeah, two to three days before, I start worrying about it. Just like the three to four hour journey getting there. The appointment itself isn't too bad because I sort of trust them.' As blood profiles normalised, these generally became less frequent, although contact with hospitals was typically maintained.

#### The continuity of significant symptoms
Clearly, symptoms reported by participants differed according to the site of their thrombosis or bleed. Participants who had experienced a stroke, for example, reported the varied long-term impact of this, with symptoms including markedly reduced mobility (one participant, for example, who was previously a club runner was limited to use of a mobility scooter when leaving home), weakness, and difficulties in speech and language. In addition, some conditions were somewhat hidden and symptom-free despite their potential for significant impact. Participants with hepatic thrombosis, for example, had hepatic hypertension and oesophageal varices requiring yearly endoscopy, but no day-to-day symptomatology.

Two symptoms, however, were common to virtually all participants: debilitating fatigue and 'brain fog' (difficulties in concentration and retention of information). Both impacted significantly on day-to-day activities, with exhaustion with usual day-to-day activities, requiring them to rest and sleep during the day. The brain fog was described by one participant: 'I can't concentrate like I used to a focus; if I were to read a book, I can't remember what I read. It's not that I've got amnesia. It's just that I can't concentrate long enough on something to focus on it and then to remember. … I used to be somebody who like, if ten people walked past me on the street, I could tell you what they were wearing, whether they had glasses… And nowadays they could be running towards me naked I may not know. So yeah, I don't have the same

retention and I don't have the same focus and attention' (M). Many continued to hope for recovery but remained disappointed: 'You're waiting for the switch to go on and all of a sudden to feel better and not knowing if you might feel should I say old self again' and continue to feel a burden to others: … 'you're so used to getting on with things so all of a sudden not to be able to do that. You know, feeling like you're becoming a burden to others, especially your loved ones' (S).

The significant continuity of symptoms meant many people struggled with day-to-day living, and many needed to rest during the busy days in order to manage their fatigue. As a consequence, social lives were curtailed and even day-to-day events within families could be stressful. In addition, a number of participants either lost their job or were able to only maintain a much restricted workload. This clearly had significant financial implications. One participant was able to quantify his restricted workload: 'I didn't want to lose my business…. We've got a big mortgage and everything, so we've got financial responsibilities. So, it was important. My main battle, while I've been recovering was to get back working. But that's not for me, normality. So normally, before I was ill, I would make maybe 100 [units] in a month. And that was what gave us a good lifestyle and a good income. And now (18 months post vaccine) I make about maybe 6 to 10 [units] a month' (F).

Alongside these continuing symptoms were exaggerated responses to more everyday physical sensations that were potentially indicative of further problems. Participants were hypervigilant and concerned when they experienced aches and pains most would ignore or simply treat with mild analgesia: 'Yeah, hypersensitive to do with my head… If I feel anything to do with my head, I do panic slightly. They just said to take a paracetamol, and if it goes away, then it can't be it….' (A).

### Lack of support and wider recognition of VITT

Many participants experienced significant financial problems as a consequence of loss of job or reduced work capacity due to continued ill health and were necessarily drawing on savings to tide them over their immediate financial crisis. Symptoms such as poor concentration and fatigue did not easily fit into the benefit system, and obtaining state benefits such as Personal Independence Payment (PIP) proved difficult and frequently went to appeal: 'We've had to fight for everything and nothing's been easy, you know? Yeah, like I said the PIP - well that was horrendous, the tribunal… Yeah, makes you feel like you're a liar' (S).

Compensation through the Vaccine Damage Payment Scheme has proven difficult to obtain, with most participants still awaiting a decision over a year after initiating a claim even in cases where there has been significant physical damage and critical levels of financial stress. This lack of financial support frequently resulted in bitterness, in that participants viewed their having the vaccination in support of a governmental policy and as a way of supporting the wider health of the population,

but the government had not reciprocated this personal commitment when people had experienced the negative effects of the vaccine, nor was it taking the plight of those affected seriously. It was also noted that the compensation available would be of limited value to individuals in their 30s who were unlikely to work again and counted against future benefit payments. Many had begun to investigate bringing legal action against the manufacturer of the vaccine in order to obtain more substantial redress. The lack of government support was an emotive issue for all, and a highly emotive issue for some: 'I get very angry about the lack of acknowledgement, support, yes. Horrific. No human being should be put through this. I think it's a stark fact, that the government produced its own VITT guidelines in which it recommended psychological support. And they can't even follow up on that. We need to rely on a charity stepping in… We really do appreciate it. We all talk, you know, we all know each other - all the names I could go through. And we are so appreciated about that because no human beings decide to go through this. I'm fully aware that if someone goes through a trauma, irrespective of whether that's imagined or real, that psychology has a massive impact. To be added on to that the burden of fighting to get support and things like that. These are the things that keep me angry. These are the things that I was always brought up not to accept …' (S).

Alongside these deficiencies in financial and healthcare provisions was a wider lack of recognition of VITT. Participants reported that governmental guidelines on control of apparently anti-vax messages had led to minimal reporting of their condition on the mainstream media, with GB News providing the only relatively consistent interest in their case, and frequent removal of social media messaging and Facebook groups that were established, as a consequence of apparently breaking rules of disinformation. Appearances of GB News provoked surprising amounts of hostility, with one participant (B) noted the comments made on one of them: 'Some of the comments were hilarious, actually. Yeah, they were. Hope you die. Somebody written, your own fault serves you right. I agree with that. The other one was repent. I got a few religious people. Yeah, you should repent. I can't… I don't think God wants to hear all that - me repenting is quite funny, actually.'

The government was seen as 'gaslighting' their experiences. As a consequence, all participants reported their condition having been denied by friends and even family, with claims that they 'really had long COVID' or simply 'COVID-19' and that they were exaggerating their symptoms: 'they just want to get off the topic – you get treated like an insane antivaxxer!' (A). Given the degree of trauma and health challenges participants experienced, the denial of their condition was particularly distressing and had on at least one occasion resulted in complete disconnection from those family members involved.

Participants were keen to make the point that they were not anti-vax. All noted that, at some time, they were clearly 'provax' in that their condition was caused by the vaccination.

Most, but not all, had declined future vaccinations, and those who did have further vaccinations were highly anxious in the surrounding days. Some were concerned about their children being vaccinated and had refused this. Most believed having the COVID-19 vaccination was a personal choice and that decisions whether to vaccinate should be made in full knowledge of its pros and cons. Indeed, one participant (M) placed herself in purdah following discharge from hospital: 'I felt like I could not talk to people about it, because I didn't want to put people off the vaccine at the time, I thought, oh, if people don't take this jab, they're going to die from COVID. So I was very careful, instead of announcing to the whole world.'

### Multiple losses

As a consequence of their continuing significant symptoms, participants reported multiple losses. Many experienced significant physical disabilities and financial losses, with basics such as mortgages and rent being put at significant risk. Alongside these losses were losses of independence, loss of future income and career, losses of social and family engagement, and loss of physical fitness and abilities. Participants also reported more subtle losses. One person, for example, noted that they struggled when visited by family members with small children as they found it mentally draining. As symptoms continued, participants reported a continuing but dwindling hope that they would be able to resume a 'normal life' (AW).

More existential losses were also evident, with participants reporting loss of self-esteem and self-worth consequent to these more obvious losses. As one previously very fit participant reported: 'It sounds a little bit arrogant if I say it now, but there was a point where I genuinely was… and this is in relation to where I am now …. Superman, whereas now I definitely feel like Clark Kent, who just, you know, he's clumsy, trips over things, he can't run. He can't, you know…' (W). F reported that: 'I'm ashamed of myself. I'm ashamed I've become this…. I was quite well known, I had a wonderful career and I miss that life … Because without that and with this illness I can't see what use I am to anyone now.' Another was more defeated in his appraisal of his life (M): 'if I went to bed and did not wake up, it wouldn't bother me… I no longer live, I just exist…' (AM). Participants had also lost a feeling of safety and felt vulnerable to both recurrence or exacerbation of their symptoms and a more general health anxiety. As A, noted, 'I didn't feel invincible (before), but I feel a lot less invincible now'. Despite the National Institute for Health and Care Excellence (NICE) guidelines[8] indicating the need for psychological support, as noted by S above, as a consequence of these multiple challenges to mental health, no participants had priority access to local mental health services.

## DISCUSSION

The likely psychological consequences of VITT have been previously considered but not reported in any depth.[9]

These data are the first to be formally evaluated and highlight the marked reduction in quality of life experienced by these individuals. As with all qualitative studies, the sample was relatively small, and has the potential to be biased, as participants were self-selected social media users and included those within a support group, which may have led to the exclusion of those with more severe injuries (and therefore not engaging in social media) and an overinclusion of those with stronger views, both positive and negative, on the impact of VITT. However, the interviews achieved thematic saturation, were reported by virtually all participants to a greater or lesser extent (usually determined by the severity and longevity of physical sequelae), and there were no discernible differences between the experiences reported by those within and without the support group. As the support group largely involved posting messages on Facebook rather than direct interpersonal meetings, the risk of data contamination as a consequence of such events is minimal.

The results highlight the challenge for governments and relevant other bodies to disseminate accurate information and encourage engagement in vaccination programmes, while simultaneously avoiding the spread of misinformation and allowing the voices of people who suffered life-changing adverse reactions from them to be heard. In addition, future mass vaccination programmes need to consider not just the benefits of any programme, but how to respond directly and immediately to individuals directly damaged by it, in a way that ameliorates rather than adds to their problems. Such responses should include meaningful and rapid financial reparation and the provision of relevant support services, both physical and psychological.

Finally, although VITT does not directly 'cause' mental health disorders, the trauma associated with its initial treatment and its longer-term impact on quality of life, through the routes identified, clearly do. Future research needs to quantify the prevalence of mental health disorders within the VITT population and further explore its impact on the patients and their partners and families.

**Contributors**  PB—interviews and lead in design, analysis and authorship of the study, acts as guarantor for the study. FC—interviews, co-analysis and contribution to authorship of the study. JW—interviews and contribution to authorship of the study. BJH—contributions to design and authorship of the study. SP—contributions to design and authorship of the study.

**Funding**  The authors have not declared a specific grant for this research from any funding agency in the public, commercial or not-for-profit sectors.

**Competing interests**  None declared.

**Patient and public involvement**  Patients and/or the public were involved in the design, or conduct, or reporting, or dissemination plans of this research. Refer to the Methods section for further details.

**Patient consent for publication**  Not required.

**Ethics approval**  This study involves human participants and was given ethical approval by the Swansea University School of Psychology Ethics Committee (approval no. 5513). Participants gave informed consent to participate in the study before taking part.

**Provenance and peer review**  Not commissioned; externally peer reviewed.

**ORCID iD**
Paul Bennett http://orcid.org/0000-0003-2252-6065

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
