## [Reviewer comments · BMJ Open]

ARTICLE DETAILS

TITLE (PROVISIONAL)	Living with vaccine-induced immune thrombocytopenia and thrombosis: a qualitative study
AUTHORS	Bennett, Paul; Celik, Filiz; Winstanley, Jenna; Hunt, Beverley; Pavord, Sue

VERSION 1 – REVIEW

REVIEWER	Moreira , Marina Beltrami Weill Cornell Medicine
REVIEW RETURNED	27-Feb-2023

GENERAL COMMENTS	Dr. Bennet and colleagues present a qualitative study investigating the acute phase and post-acute phase experiences of persons who developed VITT following adenoviral-based COVID-19 vaccines. I do not have the technical expertise to judge the methodology of this study. One obvious limitation is the sampling resulting from patients who are engaged in social media. This could create bias to include either patients who are more physically or emotionally symptomatic or conversely to exclude the most severely affected patients (affected to the point that they cannot engage in social media). A broader recruitment effort would require access to protected health information, identifying patients based on hospital diagnosis. Patients may not have provided consent to be approached about research, making this approach ethically challenging. The authors have commented on this limitation and on their efforts to address it. Despite the above, I find this study's results compelling and relevant. The patients' testimonies raise a series of systems-, culture-, and policy-related issues that are often overlooked or not emphasized enough in the clinical literature and practice. Their reports should prompt us, medical professionals, to reflect on our own practices. They highlight the importance of patient advocacy. This study also raises difficult questions about the balance between avoiding the spread of misinformation and giving voice to people who suffer very real adverse reactions from vaccinations. I suggest the authors reference more recent management recommendations (J Thromb Haemost. 2022 Jan;20(1):149-156. doi: 10.1111/jth.15572. Epub 2021 Nov 10. PMID: 34693641), which mention, albeit briefly, the importance of long-term support to persons who suffer VITT.
--

REVIEWER	Kanack , Adam J Mayo Clinic, Laboratory Medicine and Pathology
REVIEW RETURNED	19-Mar-2023

GENERAL COMMENTS	Bennett, Celik, Winstanley, Hunt, and Pavord describe VITT's physical and socioeconomic impacts on 14 interview participants
--

	who suffer from this disorder. They describe both acute and long-term trauma associated with diagnosis prior to or during the recognition of VITT as a vaccine-related thrombotic disorder. As a result, patients experienced negatively viewed diagnostic journeys, the uncertainty of clinical outcomes, trauma associated with COVID lockdowns, and challenges in maintaining social contact with family and friends in the acute time surrounding VITT development. In the long term, patients experienced difficulty reintegrating into their lives due to physical limitations caused by VITT and additional stressors associated with loss of income and self-identity. Additionally, patients felt poorly supported by a governmental health system that advocated for vaccine administration but that was later viewed as failing to recognize or support the adverse outcomes associated with VITT. Due to physical, social, financial, and associated traumas, VITT patients experience ongoing mental health disorders and seek improved recognition and support for their medical condition. These experiences also provide a framework for governmental and health organizations to recognize and implement improvements for dealing with and supporting patients with rare but genuine adverse post-vaccination reactions associated with the implementation of large-scale vaccination programs. Together, this manuscript provides a unique perspective to VITT research and insight into the patient experience with VITT.
--	---

REVIEWER	Fan, Bingwen Eugene Tan Tock Seng Hospital, Haematology
REVIEW RETURNED	21-Mar-2023

GENERAL COMMENTS	This is an important qualitative study which documents the psychosocial aspects of a life-threatening, novel, vaccine associated complication and the team should be congratulated for this. The authors present findings of a psychosocial study exploring the experiences of 14 patients with VITT in hospital and after discharge, conducted up to 18 months following their acute illness with VITT. Major points  1. How was the diagnosis of VITT confirmed/verified for each individual participant? It is understood that participants were recruited through social media via the closed VITT Facebook support group or via the thrombosis UK twitter feed or through retweets, however, many disease support groups do have participants who are actually caregivers or family members (rather than patients) who join the group to better understand the disease, to share experiences, seek avenues for therapy and to support their loved ones or others. 2. The nature of the interviews conducted over Zoom was described as “a semi-structured virtual interview”. Can the authors provide a copy of the interview questions as well as informed consent form in an appendix for reference? Did the authors also exclude significant psychiatric history (such as a background of anxiety/depression), alcohol dependency and recreational drug use, and poor socioeconomic background which may compound the themes of “struggle to optimise care” and “continuity of significant symptoms” 3. Further key demographics of the interviewees such as race, ethnicity, religion, income, education, home ownership, sexual orientation, marital status, family size, health and disability status, and psychiatric diagnosis should be furnished for readers to better understand the study population interviewed.
---

	4. The authors state that the interviews achieved thematic saturation. Which themes predominated and were the most prevalent amongst the participants? It would be helpful if the authors can come up with a summary table of the 2 main thematic issues, ie. The trauma or care, and surviving the long haul. Minor points: 1. The paper highlights significant psychological consequences of VITT with an overall negative quality of life experienced by these individuals. One silver lining amidst the gloom is the use of social media (Facebook and Twitter) as platforms for support groups. Did the researchers explore the impact of these platforms as a means of support for these survivors of VITT, where it is known that survivors from traumatic, life-threatening illness have been found to have benefit from support groups. Groves J, Cahill J, Sturmev G, Peskett M, Wade D. Patient support groups: A survey of United Kingdom practice, purpose and performance. Journal of the Intensive Care Society. 2021;22(4):300-304. doi:10.1177/1751143720952017 2. Many of the participants suffered from symptoms of anxiety and depression even after discharge (given significant losses such as loss of energy, loss of employment, loss of sense of well-being). Were these participants referred on to psychological or psychiatric services and support by the respective healthcare teams caring for them, or even by the investigating team?
--	--

VERSION 1 – AUTHOR RESPONSE

Reviewer 1

We have strengthened discussion of the limitations of the study (page 14).
We have added the suggested reference.

Reviewer 2

Reviewer 2 did not identify any suggested changes to the document.

Reviewer 3

- Additional information is given to show how participants were confirmed in their VITT diagnosis (date and hospital in which diagnosed and symptom check during interview). (page 5)
- As requested, the interview schedule is uploaded as an appendix. We have not uploaded the consent form, as this was not requested by the editor and was a fairly standard one in which participants agreed to a recorded interview and publication of anonymised quotes within a contextual passage. We would be happy to do so if required.
- We did not assess, nor exclude participants on the grounds of previous psychiatric history. We became aware of this in one interview but did not specifically check for such a previous diagnosis/engagement with psychiatric services. In retrospect, this may have been useful in terms of defining the population under study. However, we believe that restricting the study population to those without this history (given the aim of the study was to understand the experience of as wide a range of VITT patients as we could recruit, and in any population there are likely some people with previous psychiatric history) would have been deleterious to the study aims. In addition, as every participant reported struggling with their hospital experience and subsequent symptoms, and the impact of the condition on their life, we do not consider any negative impact to be a direct consequence of psychiatric risk.
- We have added additional demographic data (SES, nationality, and ethnicity). This is all the data we

have, although if more is required, we can contact participants and ask for more. (page 5)

- The reviewer asks which themes predominated and were the most prevalent among the participants. This is not an easy question to address, as the themes reported were consistently reported by almost all participants, with some variation in the nature of these accounts according to the severity of residual symptoms and speed of recovery. As such, there were no dominant themes. We have noted this in the discussion to emphasise this point (page 14)

- The reviewer makes an interesting point about the use of social media. This was raised by some participants, but not sufficiently to form a theme, apart from reporting that they were often blocked on twitter/facebook (which we do report). Interestingly, the pros and cons of social media are much more predominant in a paper exploring the experiences of families of patients with VITT, where it forms a major theme.

- We note that no participants had priority access to the local provision of psychological or psychiatric support, other than that available within the local services (and with the long waiting list this implies). (page 13). We did offer support should they feel they would benefit (see existing statement in the method section).

Finally, as noted in our original submission, we acknowledge the paper is over the recommended limit of 4,000 words, but believe significant shortening would result in a loss of information and reduce the impact of the paper. We would therefore ask that the paper is considered acceptable at its present length.

VERSION 2 – REVIEW

REVIEWER	Fan, Bingwen Eugene Tan Tock Seng Hospital, Haematology
REVIEW RETURNED	29-May-2023
GENERAL COMMENTS	The revised manuscript reads well and I congratulate the authors on their undertaking and publication of this important study.